# Waste Mineral Wool and Its Opportunities—A Review

**DOI:** 10.3390/ma14195777

**Published:** 2021-10-02

**Authors:** Zhen Shyong Yap, Nur Hafizah A. Khalid, Zaiton Haron, Azman Mohamed, Mahmood Md Tahir, Saloma Hasyim, Anis Saggaff

**Affiliations:** 1School of Civil Engineering, Faculty of Engineering, Universiti Teknologi Malaysia, Skudai 81310, Johor, Malaysia; zsyap2@graduate.utm.my (Z.S.Y.); zaitonharon@utm.my (Z.H.); azmanmohamed.kl@utm.my (A.M.); 2Centre for Advanced Composite Materials (CACM), School of Civil Engineering, Faculty of Engineering, Universiti Teknologi Malaysia, Skudai 81310, Johor, Malaysia; 3UTM Construction Research Centre, Universiti Teknologi Malaysia, Skudai 81310, Johor, Malaysia; mahmoodtahir@utm.my; 4Civil Engineering Department, Faculty of Engineering, Sriwijaya University, Kota Palembang 30128, Sumatera Selatan, Indonesia; salomaunsri@gmail.com (S.H.); anissagaf@yahoo.com (A.S.)

**Keywords:** composite behavior, material composition, materials recycling, reuse, sustainable resource, waste characterization

## Abstract

Massive waste rock wool was generated globally and it caused substantial environmental issues such as landfill and leaching. However, reviews on the recyclability of waste rock wool are scarce. Therefore, this study presents an in-depth review of the characterization and potential usability of waste rock wool. Waste rock wool can be characterized based on its physical properties, chemical composition, and types of contaminants. The review showed that waste rock wool from the manufacturing process is more workable to be recycled for further application than the post-consumer due to its high purity. It also revealed that the pre-treatment method—comminution is vital for achieving mixture homogeneity and enhancing the properties of recycled products. The potential application of waste rock wool is reviewed with key results emphasized to demonstrate the practicality and commercial viability of each option. With a high content of chemically inert compounds such as silicon dioxide (SiO_2_), calcium oxide (CaO), and aluminum oxide (Al_2_O_3_) that improve fire resistance properties, waste rock wool is mainly repurposed as fillers in composite material for construction and building materials. Furthermore, waste rock wool is potentially utilized as an oil, water pollutant, and gas absorbent. To sum up, waste rock wool could be feasibly recycled as a composite material enhancer and utilized as an absorbent for a greener environment.

## 1. Introduction

The volume of global solid waste generation is estimated at 1.3 billion tons per year [1] and it is expected to climb to about 70%—2.2 billion tons by 2025 [2,3]. One of the major global solid wastes is mineral wool waste which is generated at 2.54 million tons per year [4]. This volume is measured based on rock wool productivity over a 30-year lifespan [5] and assuming 5% wastage during installation [6]. The volume of rock wool was estimated to have grown linearly from 2.25 million in 2010 to 2.54 million in 2020 [5]. It is expected to grow to 2.82 million in 2030, as shown in Figure 1 based on the researchers’ calculation. It is notable that an increasingly massive amount of fibrous waste will become a pinnacle issue to the environment [7,8] and economy [9,10] if left unresolved for a long period.

Rock wool or stone wool is the main mineral wool with fine and intertwined fibers produced by spinning molten rocks at high speed that is akin to make cotton candy. The mass of fibers is shaped into sheets with two prominent characteristics—low density and high porosity, making rock wool an excellent construction choice for sound absorption, thermal insulation [11], and fire resistance [12,13]. In the light of these prominent characteristics, rapid development and industrialization of rock wool insulation material started in the early 20th century [14]. To date, rock wool has become commonplace as a building material [15] and pipe insulation material [16] where it makes up more than 50% of the world’s insulation material in the market [17,18]. Alongside this development is the massive amount of degraded rock wool. It is generated and discarded as waste [19].

The landfill and dumping areas are practically the most common solutions to waste rock wool. However, it comes with a series of common environmental issues [20,21,22], and most are connected to greenhouse-gas emissions [23,24,25]. The second major problem concerns the larger space and logistics arrangement required to store the loose and bulky waste rock wool. This issue further aggravates the difficulties related to landfill shortages [26], natural resources shortages [27,28], and an increase in waste disposal costs [29,30]. These remain a crucial issue for the wool fiber industry throughout the world.

Several studies have been published to overcome the conundrums on the reclamation of waste rock wool into the rock wool production process [31,32]. It encompasses introducing waste rock wool as recycled fillers into different materials, including pozzolanic material [33], geopolymer concrete [34], alkali-activated material [35], polymer composites [36], and wood-fiber composites [37] as well as reutilizing the waste as a pollutant sorbent to remove split oil [38] and hexavalent chromium Cr(VI) [39] from water and filter hydrogen sulfide (H_2_S) gas. However, none has performed a thorough review of comparing the available recycling techniques of waste rock wool to understand its feasibility for the above functions.

As such, this paper reviews the available recycling methods and the inherent properties of rock wool before concluding with some recommendations for future research development. Hence, the objectives of this review are: (i) review the characterizations such as physical characterization, chemical composition, and possibly contaminants of waste rock wool, (ii) review both the advantages and disadvantages on the performance of the waste rock wool reinforced composites as well as other recycling approaches, and (iii) propose critical perspectives or knowledge gaps for future research. With much confidence, this review would benefit both academia and industry to formulate more scientific and practical strategies to handle waste rock wool in the future.

## 2. Research Approach

This review takes a comprehensive approach to analyzing the research outputs in the field of waste rock wool that have been published in Scopus, ScienceDirect, and Google Scholar, which are the widely used search engines for academic outputs. The keywords used in the literature search were “rock*wool” or “mineral wool” or “man-made mineral fiber” or “man-made vitreous fiber”, as well as other related terms such as “waste*” or “recycle” or “used”. These keywords were applied with Boolean operators, quotation marks, wildcards, and query sets.

The complete workflow is shown in Figure 2, which includes the literature searching and screening procedures, and the scope of the qualitative discussion. As illustrated in Figure 2, the literature search was divided into three sub-steps to weed out articles that were either out of scope or did not focus on the recycling of waste rock wool. The literature was manually screened by first reading the abstract and findings.

Following literature searching and screening, a qualitative discussion was conducted. After reviewing the finalized literature, the review structure was categorized into three main groups for in-depth qualitative discussion. The three important scopes are:Identify the characterization of waste rock wool (physical properties, chemical composition, and contaminants).Review the performance of composites after incorporated waste rock wool.Investigate other applications and their effectiveness.

## 3. Waste Characterizations

Waste rock wool is commonly generated from two primary sources: manufacturing and post-consumer waste [40,41]. The characteristics and volume of waste from both sources are significantly different. Manufacturing waste is cleaner with a known composition that is commonly reintroduced into its manufacturing process. This method is an effective way to lower the raw material cost by replacing coke usage. Moreover, a recent study compared the melting behavior of virgin raw materials and waste rock wool and concluded that re-melting the waste rock wool is 23% more energy-efficient on differential scanning calorimetry (DSC) [31].

For post-consumer waste, a study pointed out that the inappropriate sorting of used waste rock wool gives rise to unknown quality and contaminations [42]. Thus, post-consumer rock wool is hard to reintroduce as raw material to its manufacturing production. This section aims to identify and quantify predominantly the physical characteristics and chemical composition present in post-consumer wastes [43] before an appropriate treatment method can be assigned.

### 3.1. Physical Characteristics

Waste rock wool has a fibrous-open porous structure and may differ in densities and strengths [44] due to degradation when exposed to moisture in the air or acid rain [19]. It is generally regarded as unrecyclable [5,34] until modern technologies have made its blending with other materials possible.

Comminution is currently the preferred method to produce a homogeneous mixture of particles [45]. It involves crushing, grinding, cutting, or other similar reduction methods to break a solid material into smaller sizes to mix with other composites. Depending on the actual process adopted, the processed rock wool’s bulk density, fiber length, and width would change [46]. Table 1 summarises the different approaches that have been adopted to produce homogenous fiber particles.

### 3.2. Chemical Composition

The chemical composition of rock wool is highly dependent on its manufacturing process. Each rock wool factory uses different raw materials based on their preference and intended use. Nevertheless, the raw material is typically from the rock crystalline group and may include anorthosite, dolomite, diabase, olivine sand, or even slag [31]. Table 2 shows the compositions of chemicals of ordinary waste rock wool from different research studies.

A general categorization of rock wool properties may seem almost impossible. However, a closer look at Table 2 would reveal that most waste rock wool consists of three major components—silicon dioxide (SiO_2_), calcium oxide (CaO), and aluminum oxide (Al_2_O_3_). SiO_2_ is the major component that accounts for around 38.7–60.1% of the total weight in most cases except in the two studies [47,51]. For CaO and Al_2_O_3_, the majority weight ratio may vary from 10 to 46.9% and 0.8 to 18.6%, respectively.

The difference in chemical composition is also apparent between industrial and construction usage based on the content, contamination level, and purity level. For instance, the rock wool used as piping insulation is often mingled with water, oil, or an unknown chemical. The wire mesh inside the rock wool panel used for pipes tends to rust over time due to water or moisture in the air. These contaminants or dirt in the porous rock wool pose the biggest hurdle to recycling efforts, making it the most apparent reason behind million tons of waste rock wool piling up every year in landfills and adversely affecting the environment. All in all, to diminish the contaminants of the waste, the initial stage of source separation is essential to getting the waste recyclable [53].

A point worth noting is that no research has performed a physical measurement on either post-consumer waste or manufacturing waste. This problem may reveal a vast difference in waste quality and its effect on the choice of recycling methods and the resultant product.

### 3.3. Contaminants

Waste rock wool is contaminated in two phases. The first phase is during the manufacturing process, where hazardous materials such as phenol-formaldehyde binders are coated onto the fiber surface [54]. The second phase is during the application process, where the waste rock wool becomes contaminated with other pollutants such as mycological substances, impurity dust, or chemical substances. With the contaminants, waste rock wool’s composition becomes complex and more difficult to be recycled [55].

The resin application as a binder during the production process is unavoidable as it is required to bind the rock wool fiber together after the molten rock fiber has been cooled down by air [56]. The amount of resin is small, typically in 1:10 wt% of the rock wool, but it can affect the properties of waste rock wool and increase the difficulty of recycling waste rock wool [57]. Another common contaminant is mycological substances, for example, fungi, present in a humid environment [58]. Fortunately, recent research shows that resin or other organic matter can be burnt off by heating the waste to a specific temperature [31]. This finding also infers that if the resin is the only contaminant present, the chance to recycle the waste rock wool as raw material is high. However, in post-consumer waste, the resin should be neutralized using hydrogen peroxide, some catalysts, and a mixture of diammonium hydro phosphate and carbamide [59].

Apart from organic contaminants, other impurities such as dust or chloride are commonly diffused in the air and easily found on the surface or even in the porous structure of waste rock wool [60]. Chloride stored in the rock wool pores forms hydrochloric acid, which is a corrosive and volatile liquid. The transformation of chloride particles to acid usually occurs when the rock wool is hot and humid. This problem can be resolved by washing ground waste rock wool with tap water at a water-to-solid ratio of 10 to significantly reduce chlorine content [51]. In the study, the chloride content was reduced from 4.901 to 0.612 wt% after stirring the waste in water for a few minutes.

Overall, the characterization of waste rock wool in terms of the physical, chemical composition, and contaminants reveals that waste rock wool from the manufacturing process has a higher potential to be recycled than the mineral waste rock wool from other industries. This is attributed to the known composition and purity content that is easier for recycling processes. Among the recycling methods, “comminution” is preferred as its recycled product has better mechanical strength and chemical resistance for application. Meanwhile, the aim of reducing contaminants can be achieved by using “washing” with water, mainly to reduce the chlorine content that is undesirable for incorporating into composites. The feasibility and cost-effectiveness of pre-treatment and recycling methods are one of the major concerns in reprocess waste. However, a thorough and systematic investigation of these two aspects is yet to be conducted. Therefore, more studies on the practicability and costing should be done to develop efficient and cost-effective waste material pre-treatment and recycling methods.

## 4. Repurposing as Fillers in Composites

Repurposing waste materials in composites are the common trend of utilizing waste and other materials to increase the property of the composites [61]. This section reviews the five potential solutions for waste rock wool to mix with other materials to form a better performance composite.

### 4.1. Pozzolanic Material

Pozzolanic material is widely used in concrete making to reduce cement content; improve the workability of fresh concrete; increase the concrete strength; enhance the durability of hardened concrete [62]; and lower the overall production cost and impact on the environment. In studies of pozzolanic material, the incorporation of waste material to drive down overall production costs [63,64] is an exciting topic with studies already done extensively on silica fume, fly ash, ground granulated blast slag [65], rice husk ash, textile fiber [66], and palm oil fuel ash [67]. Many continued to perform with a proven track record in industrial applications [68,69,70]. The study on rock wool is considered a more recent venture to ultimately achieve the same objective [71,72].

The significant enablers for rock wool to become a pozzolanic material are the high SiO_2_ (up to 70%) and CaO (up to 20%) content. In the amorphous phase, silica is necessary to initiate and maintain pozzolanic activity. Since the pozzolan reaction only occurs on the silica particle surface, the specific surface area is the most significant factor in the effectiveness of pozzolanic activity [71,72,73]. A study has been conducted on fine rock wool particles to facilitate the early hydration of cement, however, the result showed a prolonged dormant period that is undesirable [51]. Table 3 below summarises other methods used to mill the thin and soft rock wool fiber with their respective composition and its incorporation into concrete-making.

One of the studies replaced 20% of cement with waste rock wool and successfully obtained a high increment in fresh concrete flow spread [71]. The ground waste rock wool had also prolonged the induction period of cement hydration at 5 wt% (1 h 17 min), 15 wt% (4 h 34 min), and 20 wt% (6 h 27 min). Beyond the 20% replacement of cement with waste rock wool, the flow spread decreased due to an unfavorable demand for water to moisten the particles’ surface [76].

In hardened concrete, waste rock wool’s fibrous nature had shown to slightly increase the porosity and decrease the density (−0.09 g/cm^3^) of the final product [33]. These could be promising to improve the sound [78] and thermal insulating properties [79]. However, in terms of mechanical properties, the observed outcomes vary for different researchers. Some studies found that the inclusion of waste rock wool had caused a little decrease in the mechanical properties of concrete [15,33,75]. However, there are some studies that reported a growth in overall strength [49,72]. The latter also reported enhancement in other properties such as compressive strength, splitting tensile strength, absorption, resistivity, chloride-ion penetration resistance, and abrasion resistance. At 50% replacement of aggregates with waste rock wool, the compression strength of concrete reduced while the flexural strength was enhanced up to 12% [77].

The contradictory findings have been addressed [51], whereby the paper attributes the differences to the presence of contaminants commonly found in waste rock wool, namely halite (NaCl) and sylvite (KCl) [33]. Without proper cleaning of waste rock wool, the NaCl and KCl content might have increased the early hydration of cement and promoted the generation of a large amount of Friedel’s salt that would consume the portlandite (Ca(OH)_2_) in cement and inhibit C_3_H gel formation needed for the development of compressive strength. Consistent with this inference, the usage of waste rock wool with negligible NaCl and KCl content would mean a positive increase in the mechanical properties of concrete [49,51,72]. It should also be highlighted that these three studies documented the washing of waste rock wool with tap water to leach off and reduce the overall chloride content (from 4.901 wt% to 0.612 wt%) before mixing with cement.

### 4.2. Geopolymer Precursor Material

The geopolymer precursor material may be an appropriate replacement to the traditional Portland cement [80,81]. It contains geological source material with silicon and aluminum content and a caustic activator to form a binder [82,83]. In 2017, a study revealed that waste rock wool is an ideal replacement for slag and fly ash from iron and metal to produce geopolymer concrete due to its high silicon and aluminum content [34]. This finding was confirmed using the selective solubility tests at pH 11.6 [84,85], which revealed that the selective solubility of milled waste rock wool (SiO_2_—73.0%, Al_2_O_3_—87.3%) was almost twice higher than fly ash’s (SiO_2_—32.9.0%, Al_2_O_3_—35.8%) [34].

After incorporating waste rock wool, the mechanical properties of geopolymer concrete showed that, with 33% rock wool and 47% fly ash inclusion in geopolymer concrete, the compressive strength after 28 days was 12 MPa [34]. In one of the studies [4], both concrete compressive and flexural strengths had increased 30 and 20.1 MPa, respectively, after the inclusion of waste rock wool. Recently, a study compared the new and waste rock wool and revealed that both geopolymer precursor materials achieve 15–20 MPa compressive strength after 1 day of curing, and resulted in 44 and 36 MPa for new and waste rock wool, respectively, after 28 days [37]. The compressive strength difference might be due to the impurities or microorganisms in the waste rock wool, as previously stated. Overall, the adoption of waste rock wool as a geopolymer shows the potential to replace cement with more sustainable material.

### 4.3. Alkali-Activated Material

The alkali activation method enables industrial by-products to be recycled [86]. Waste rock wool contains a high amount of SiO_2_, Al_2_O_3,_ and CaO which thus are suitable as the precursor for producing alkali-activated materials [87,88].

Attempts to establish alkali-activated material as a more environmental alternative to Portland cement [89] are not new. In fact, some had shown promising performance in the enhancement of early and late strength development [90,91], high-temperature resistance [92,93], durability under chemical attack [94,95,96] as well as adhesion to metallic and non-metallic surfaces [97]. Alkali-activated material is identical to geopolymer but different in the chemical reaction. It has lower calcium content than geopolymer, higher mechanical strength but is not as durable.

Milled waste rock wool is a promising precursor to produce alkali-activated cementitious binder because of its favorable chemical composition and high surface area [35,98]. The highest compressive and flexural strength is currently recorded at 30.0 and 20.1 MPa, respectively [4]. The activation process in waste rock wool with cement starts when Mg-O, Ca-O, Si-O-Si, Al-O-Si, and Al-O-Al bonds are destroyed, and a layer of Si-Al is formed on the grain’s surface to form hydration products [88]. The rock wool-based alkali-activated binder comprises an amorphous sodium and aluminum substituted calcium silicate hydrate (C-(N-)A-S-H) gel, the main constituent in concrete. This gel increases when Na_2_O.2SiO_2_ is used as the alkali activator to dissolve more Si [99]. Increased soluble Si content generally increases the rock wool-based alkali-activated binder’s density to allow more aluminosilicate chain-crosslinking in the C-(N-)A-S-H-type gel and the creation of an extra Al-rich N-A-S-H gel product during the N-A-S-H gel reaction [100]. The major challenge lies in controlling efflorescence formation during concrete mix design [101].

### 4.4. Polymer Composites

A polymer composite is a multi-phase material with its polymer matrix integrated with reinforcing fillers to enhance the final product’s mechanical properties [102,103]. The incorporation of low-cost recycled fillers such as fiber [104,105], carbon black, and silica in rubber [106,107] into the polymer is currently a hot research topic of reducing conventional fillers’ usage. For waste rock wool, a study depicted agglomerated milled waste rock wool (with diameter 1–3 µm and length 8–200 µm) as a particular filler into a different type of polyethylene polymer: high-density polyethylene, low-density polyethylene, recycled high-density polyethylene, and recycled low-density polyethylene [108].

Polyethene is a high-flammable material. By introducing waste rock wool as filler to the polyethylene, the polyethylene ignition time could increase 6–8 s, and the mass loss of the composite after heated could be halved [108]. This performance indicates that the developed composites are less liable to combust and is, in fact, a commercially attractive property to produce high-temperature resistant products.

The introduction of 40% weight of waste rock wool into the polyethylene has shown an increase in bending stress by 5 to 6 MPa, impact energy by 5 to 7 kJ/m^2^, and Brinell hardness by 12 to 18 MPa alongside a decrease in tensile stress by 10 to 15 MPa for all types of polyethylene. Composites with virgin rock wool fillers performed better than those containing recycled rock wool. This result could be explained by the reinforcing influence of the activated non-hardened resin.

However, another study recorded a significant drop of 93% in impact strength for the composite with 40 to 60 % waste rock wool fillers [36] similar to the study on recycled wood-polyurethane composite [109]. The composite’s Brinell hardness was not increased by the addition of waste rock wool fillers even up to 60%. The authors concluded that incorporating micro-fillers into polymer had increased the porosity of the composite beyond favorable conditions. A point to note is that the study did not mention the fineness of the fillers. It also included another variation of the study, which was adding 3% coupling agent and 3% of processing additive. The effect of these variations on the overall result remained unknown [109]. As such, it was stating that waste rock wool is not an effective filler may not be an objective conclusion in this case.

Undoubtedly, more research on the incorporation of waste rock wool into polyethylene is required to validate the properties of the resultant polymer composites. A more standard approach is also required and with due consideration given to the approach’s economics and carbon footprint to understand its applicability better.

### 4.5. Wood-Based Composites

The potential of waste rock wool in wood-based composites is worth highlighting through the study, incorporating the waste into particleboards to reduce ignitability [110]. The study showed 20 to 30% addition of rock wool had shortened the smoldering path by 19 to 30%. Although this reduction in ignitability is desirable, it was accompanied by a deterioration in the modulus of rupture (up to 65%) and internal bonding (up to 71%) due to the low cohesion of the rock wool domains within the board.

A study noted a remarkable increase in durability performance after adding rock wool filler into a fiber composite [111]. This result is attributed to the high tensile strength, modulus, chemical resistance, and dimensional stability of the rock wool fiber mineral [112]. Profound enhancement of thermal insulation was also noted [37] and moisture resistance properties [111]. However, there was still a decrease in mechanical properties.

For plaster composite, the reinforcement of recycled rock wool at a maximum percentage of 10% (in weight) and a waste-plaster ratio of 0.6 to 0.8 [15] had enhanced the flexural strength by 26.58% [15]. This finding was slightly higher than the results obtained in previous studies pertaining to plaster or gypsum reinforced with short sisal fibers [113]. Moreover, adding 4% of waste rock wool enhanced 14.64% of the surface hardness—a feature essential to reduce the defects induced during the transportation and installation processes. However, the plaster composite’s compressive strength had decreased due to the increment of the pore in the material that had also increased its density by 6.75%.

Current studies have generally depicted a reasonable enhancement in the ignitability and durability of wood-based material incorporated with rock wool fillers, but deterioration of mechanical properties cannot be neglected. As such, future studies in this aspect are inevitable.

## 5. Other Applications

### 5.1. Oil Absorbent

Porous materials can be used to absorb oil contaminants from water [38]. Common sorbent materials include birch bark, glass wool, cork, polyurethane foam [114], aerogels, and polystyrene [115]. The major problem with these absorbents is the limited capability to separate high-viscosity oils. Waste rock wool, in this case, has recently been reported to have overcome this limitation, providing a lower cost and environment-friendly alternative [38].

Fibers’ efficiency in reclaiming water contaminated with crude oil is dependent on their surface qualities [116]. Waste rock wool is associated with oil and water due to its functional group (-OH groups on the surface). To turn waste rock wool into an effective oil sorbent, a study adopted the dip-coating method (PDMS/SiO_2_ nanoparticle mixture) to turn the waste rock wool into superhydrophobic material [38]. With a thin layer coating of PDMS and SiO_2_ nanoparticles on rock wool fiber, as illustrated in Figure 3, the superhydrophobic properties over a water contact angle from 0 to 152.9° have been enhanced, which is excellent in repelling water.

In terms of absorption effectiveness, a kilogram of waste rock wool can absorb 8 to 13 kg of spilled oil, depending on the oil density. The modified rock wool’s volume absorption ability could reach up to 90% [38]. Furthermore, waste rock wool may absorb spilled oil in a circular fashion and reload after combustion or squeezing without deforming its shape, which is advantageous for transportation and post-treatment activities. Due to its inorganic composition, it is also mechanically stronger than other absorbents.

The most notable property of rock wool is its exceptional thermal stability. It allows application in an extremely hot environment, indicating that high viscosity oil absorption continuity during combustion is possible.

### 5.2. Hexavalent Chromium Cr(VI) Absorbent

Another study has succeeded in modifying waste rock wool by coating zero-valent iron (ZVI) nanoparticles onto its fiber surface to remove hexavalent chromium, Cr(VI), from water through adsorption and reduction [39]. Cr(VI) is a form metallic element that enters drinking water sources via dye discharges and paint pigments [117], wood preservatives, chrome plating wastes, and leaching from hazardous waste sites [118]. In this case, the waste rock wool was modified using hydrochloric acid to load more Zero-valent Iron (ZVI) nanoparticles onto the surface (Figure 3). Due to the steric hindrance effect, acid-modified waste rock wool could effectively limit the aggregation and oxidation of ZVI nanoparticles and successfully removed 197.69 mg/g of Cr(VI) in 30 min.

By comparing with other trendy absorbents such as bamboo biochar with a maximum absorption capacity of 61.88 mg/g [119], ethylenediaminetetraacetic acid—chitosan modified Fe_3_O_4_@SiO_2_ (47.27 mg/g) [120], and diethylenediamine functionalized magnetic carbon-based adsorbents (22.24 mg/g) [121], the acid-modified waste rock wool has the most outstanding absorption performance. In conclusion, modified waste rock wool has great potential as an absorbent to remove Cr(VI); however, the removing efficiency on higher concentration pollutants could be investigated in future studies.

### 5.3. Hydrogen Sulfide (H2S) Gas Absorbent

For gas absorbance, at the early stage, a few studies discovered rock wool as a gas absorbent to remove wastes such as polycyclic aromatic hydrocarbons [122] and sulphur gas [123]. Recently, a few studies successfully established waste rock wool as a promising absorbent for toxic and corrosive biogas—hydrogen sulfide (H_2_S) gas [124,125,126]. Biogas usually contains carbon dioxide, methane, and a small amount of hydrogen sulfide, which people usually utilize to generate energy. However, toxic H_2_S gas will cause damage or corrosion to the combustion engines. Hence, contaminants in biogas, particularly hydrogen sulfide, must be eliminated [127].

The waste rock wool was added into rod mill waste at a ratio of 19:1, designed as a vertical perforated gas tube (Figure 3). H_2_S gas was administered with a downward flow, and the rock wool filter succeeded in removing 87.9% [128] and 98% of the sulfide gas over 80 days. Another study emphasized that hydrogen sulfide gas’s actual gas flow at the landfill is different from the study [129]. Because of this, the actual scale of H_2_S gas elimination should be studied further in actual condition as well as its efficiency as compared to typical approaches such as electrochemical precipitation [130], ion exchange [131], membrane separation [132], and typical adsorption [133,134].

## 6. Conclusions

This paper has broadly reviewed the characteristics of waste rock wool, its recycling methods, and finally, its potential usages based on various studies. Several remarks are concluded as follow:Waste rock wool constitutes a high content of SiO_2_ (38.7–60.1%), CaO (10 to 46.9%), and Al_2_O_3_ (0.8 to 18.6%) generally. This contributes to the enhancement of cementitious properties such as strength, chemical resistivity, and durability. Therefore, waste rock wool is regarded as a strong potential alternative to fillers in primarily concrete production.Incorporating waste rock wool into wood-based composites resulted in an increase in ignitability and durability, but deterioration in mechanical properties.The performance of composite materials after being incorporated with milled waste rock wool is disputable. This is attributed to the presence of various contaminants such as resin, dust, chemical substances, and mycological substances, depending on their source and handling method. It has posed a significant hurdle to formulate a standardized method to treat or recycle waste rock wool.Various pre-treating and reprocessing strategies such as heating, washing, and comminution are adopted to maximize the usability of waste rock wool. Heating can burn off the resin while washing could significantly reduce chlorine content. Comminution can produce a homogeneous mixture of particles that allow waste rock wool to be incorporated into different types of composites to improve desired properties.Waste rock wool is also promisingly utilized as an absorbent for spilled oil, hexavalent chromium contaminants, and hydrogen sulfide (H_2_S) gas. The high heat stability and inorganic composition of rock wool are its notable features since it is mechanically stronger and may be used in high-temperature environments. However, the pre-treatments such as PDMS/SiO_2_ nanoparticle dip-coating and Zero-valent Iron (ZVI) nanoparticles coating on the fiber surface are needed, which are neither energy nor cost-effective to the recycling process.

Results of the pertaining studies are more consistent in this regard and are worth exploring beyond current development. Nevertheless, comparison with other existing methods is encouraged to ensure its commercial viability and applicability in the field beyond laboratory setup.

## 7. Key Perspectives for Future Research Development

Generally, research on fibrous wastes is not as profound as other waste materials. To be more efficient and commercially viable in the recycling, repurposing, and reusing of waste rock wool, the following are suggested for future development:

Purity classification of waste rock wool from different sources needs to be performed prior to determining the suitable approach to handle the waste. This classification should primarily address the type of rock wool, source, prior usage, types of contaminants, physical condition, and chemical composition.

The current comminution process is neither energy nor cost-effective to pre-treat waste rock wool. This process is seen as the critical factor in driving down the overall cost and sparking more interest in research and development.

## Figures and Tables

**Figure 1 materials-14-05777-f001:**
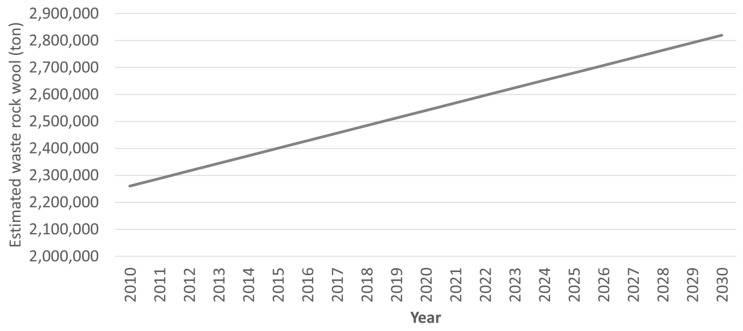
Annual waste mineral wool estimation in European Union Countries. (Modified from [5]).

**Figure 2 materials-14-05777-f002:**
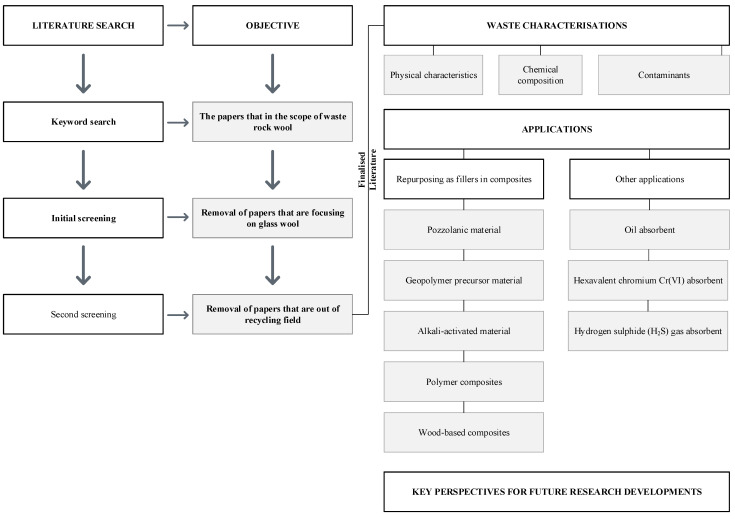
Literature search and scopes of the review.

**Figure 3 materials-14-05777-f003:**
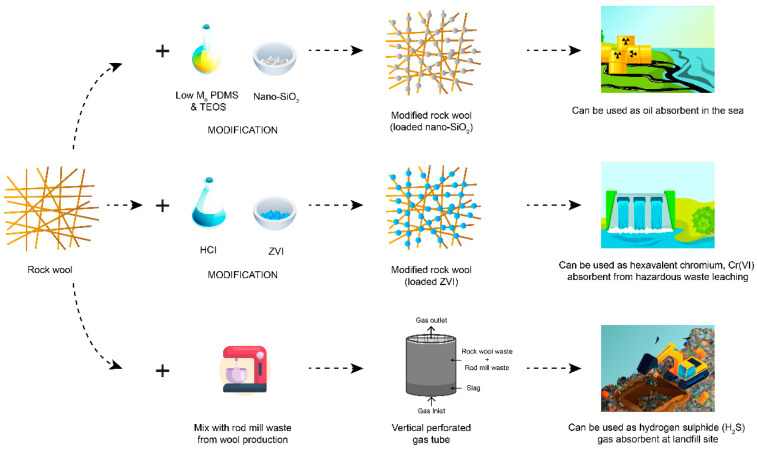
The applications of waste rock wool and its modification methods.

**Table 1 materials-14-05777-t001:** The different approaches to comminution for waste rock wool.

Machine Used	Average Fibre Length (µm)	Average Fibre Width (µm)	Average Particle Length Versus Width	Percentage of Particles <20 µm (%)	Citation
Raw rock wool	424	7.6	55.7	1.2	[46]
Granulator 200 series	387	7.6	50.6	1.2	
100 UPZ fine-impact mill	365	8.1	45.0	1.5	
ZRI homogenizer	191	7.7	24.7	2.0	
Hydraulic press 30 ton	76	10.3	7.4	3.0	
Vibratory disc mill RS200	37	11.7	3.2	19.0	
	33.6	-	-	-	[4]
Laboratory scale Agitator	4760	500	9.5	-	[47]

**Table 2 materials-14-05777-t002:** Composition of chemicals of waste rock wool from different studies.

Composition (%)									Sources
SiO_2_	CaO	Al_2_O_3_	Fe_2_O_3_	MgO	K_2_O	Na_2_O	SO_3_	TiO_2_	
38.70	20.90	18.60	5.30	7.00	2.00 *		NA	NA	[48]
16.90	46.90	5.40	16.20	2.60	NA	NA	NA	NA	[47]
40.6	3.52	2.14	6.91	11.1	6.34	6.71	2.41	0.23	[49]
40.40	17.40	1.80	9.20	12.60	0.40	NA	NA	0.80	[4]
42.00	14.70	16.60	11.30	12.20	0.50	1.60	0.03	0.90	[34]
44.06	16.36	15.94	11.93	5.68	0.57	3.71	NA	1.58	[50]
27.04	9.57	6.31	8.03	5.85	5.45	3.43	5.05	NA	[51]
40.76	21.09	11.48	11.30	NA	NA	NA	NA	NA	[52]
60.10	22.60	0.76	1.36	1.39	0.58	5.29	1.36	0.20	[33]
42.60	18.30	18.10	7.10	8.40	0.60	2.10	NA	0.80	[31]
40.00	14.50	16.00	9.00	14.50	2.00	NA	NA	NA	[46]
44.1	16.6	14.3	5.5	14.7	0.3	1.2	0.0	0.2	[37]
51.54	13.94	1.07	0.36	2.92	0.56	8.20	0.3	0.04	[42]
40–52	10–12	8–13	5.5–6.5	8–15	0.8–2.0	0.8–3.3	0–0.2	1.5–2.7	[44]

* the summation of K_2_O and Na_2_O.

**Table 3 materials-14-05777-t003:** Physical and chemical characteristics, and mixing method of incorporating waste rock wool into concrete.

Fiber Length (μm)	Density (kg/m^3^)	Major Chemical Composition	Mixing Methods	Mixing Proportion	Sources
75	206	SiO_2_ (38.7 wt%) CaO (20.9 wt%)	Replace cement	10%, 20%, 30% & 40% (mass)	[74]
24.4	-	SiO_2_ (40.6 wt%)	Additive	1%, 2% and 3%	[49]
-	-	-	Replace aggregate	30% (mass)	[75]
1–50	400	SiO_2_ (30.0 wt%) CaO (10.35 wt%)	Additive	5%, 15% and 20%	[51]
45	742	SiO_2_ (53.51 wt%) CaO (22.5 wt%)	Replace cement	25% (mass)	[76]
500–1000 μm	-	SiO_2_ (70.0 wt%) CaO (20%)	Replace sand	30% (mass)	[33]
500–1000 μm	-	-	Replace aggregates	30%, 40%, 50% (mass)	[77]

## Data Availability

Not applicable.

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
