# Peer review of "Waste Mineral Wool and Its Opportunities—A Review"

_materials, 2021, doi:10.3390/ma14195777_

Round 1
Reviewer 1 Report
Dear authors
the review, in general, is well written, interesting well discussed, with scientific soundness, you used a great quantity of information, all figures are clear and explanatory. I have no problem recommending this review for publications
Minor comments:
a) It seems that you forgot to remove the "Introductions instructions of the template" please remove (Pag. 78-87)
b) Do not use future auxiliary words or verbs in the objective of the papers such as "will" (Pag. 69)
c) Please separate numerical data value from its units (Example; Pag. 223, 224)

Author Response
Dear Honorable Reviewer,
Thank you for the useful and impactful reviews. The new version of manuscript is marked up using the “Track Changes” function in the attached MS Word file. Here we would like to humbly highlight the actions addressed based on the comments and suggestions :
a) It seems that you forgot to remove the "Introductions instructions of the template" please remove (Pag. 78-87)
Response: Thank you for the kind notification. In the new version of article, we have removed the redundant part as shown from Line 95-104.
b) Do not use future auxiliary words or verbs in the objective of the papers such as "will" (Pag. 69)
Response: The amendment has been made as shown in the Line 86. Furthermore, the overall manuscript was checked to ensure no future auxiliary words or verbs. Thank you.
c) Please separate numerical data value from its units (Example; Pag. 223, 224)
Response: The numerical data value from its units was separated as shown in the Line 261, 262. The overall manuscript was checked also and confirmed that there is no similar mistake.
Again, sincerely thanks to the Reviewer for the valuable comments and time.

Reviewer 2 Report
The authors written a detailed and very interesting review about the huge potential of using rock wool waste. The introduction highlights the characteristics that make rock wool waste very attractive: low density and high porosity. Moreover, the authors addressed critical points (such as: the chemical composition and the presence of contaminants in rock wool waste) that turns the recycling of this material more difficult. Also, the authors highlighted the impact of the manufacturing process, as well as, of the cleaning process on the chemical composition and number of contaminants. Lately, the authors showed that the inclusion of rock wool waste in composites increases the porosity, durability and confer heat-resistance to these materials.Another attractive,cost-effective and eco-friendly application is using rock wool waste as an oil-absorbent.
The authors did a great work and so, I recommend accepting the manuscript in the present form.
Author Response
Dear Honorable Reviewer,
Thank you for the kind words and valuable time in reviewing our work. We really appreciate it. Hope this review study could benefit our readers in the future. We hereby attached the new version of manuscript with the corrections based on the comments from other reviewers. The new version of manuscript is marked up using the “Track Changes” function in the attached MS Word file.

Reviewer 3 Report
The paper is a very interesting work on the reuse of waste mineral wool as a secondary raw material, increasing knowledge on the subject and raising awareness on issue and opportunity of reusing this material. In my opinion it is worthy of publication.
Minor revision:
Figure 1 (page 2): better specify the source of the data in the figure
Author Response
Dear Honorable Reviewer,
Thank you for the useful and impactful reviews. The caption for Figure. 1 has updated with the source of the data. The new version of manuscript is marked up using the “Track Changes” function in the attached MS Word file.
Again, sincerely thanks to the Reviewer for the valuable comments and time.

Reviewer 4 Report
Dear Authors,
Thank you for your manuscript, here will be following comments:
- The referenced sources of your literature are rather limited and mostly between 2014-2021, in general you cover years 2010-2021. Why? Why not to cover 20, 30 years? This material was quite commonly used before 2010 too. Lines 89-90: “…included an extensive search for papers dealing with waste rock wool” if you mention this then your references list is not sufficient. Please elaborate extensive search of papers and add some valuable references to your list from before 2010 too.
Your format indication of references in the text is not constant: first you number it (1,2) then you (Author, year)! Why? Please check if you have correct spelling and order of authors names in your references, one of the examples is Ref. 48 – wrong author order.
- Abstract doesn’t indicate what is novel in your review, why did you wrote it and what you would like to address to a reader.
- Lines 60-61, if you mention several studies, please add also some references, not only one.
- Lines 74-87: you have to remove those lines! That is from MDPI template!
- The research approach is rather poor described, please check several review papers published lately where they indicated a more detailed info how to performed structured literature search. In your case lines 89-97 are unnecessary, though figure 2 can remain and be updated. It would be more important that you indicate in the beginning of your review in Introduction “WHY did you perform this review and some bullet points indicating it” and based on that structure and novelty to organize your paper.
- Conclusions have to be elaborated.
Is your paper an extraction of info from Bachelor or Master thesis?
Please elaborate it properly, for now the decision is "major" in to side of "Reject"
Author Response
Dear Honorable Reviewer,
Thank you for the useful and impactful reviews. The new version of manuscript is marked up using the “Track Changes” function in the attached MS Word file. Here we would like to humbly highlight the actions addressed based on the comments and suggestions:
- The referenced sources of your literature are rather limited and mostly between 2014-2021, in general you cover years 2010-2021. Why? Why not to cover 20, 30 years? This material was quite commonly used before 2010 too. Lines 89-90: “…included an extensive search for papers dealing with waste rock wool” if you mention this then your references list is not sufficient. Please elaborate extensive search of papers and add some valuable references to your list from before 2010 too.
Response: Thank you for the comment. We have included the early year of the references. Here are the summary of the citation:
2014: [5], [16], [59], [96], [100], [111], [129]
2013: [6], [14], [19], [41], [48], [71], [74], [89]
2011: [72], [87], [110]
2010: [7]
2009: [32], [60]
2006: [18], [21], [123]
2001: [20], [113]
1989: [122]
Total 27 references for the early stage were included. It is true that rock wool was commonly used for buildings and industries, however, not many studies were done on the investigation of recycling or reusing methods for this post-consumer waste material at the early year. Therefore, most of the discussed references are between 2014 and 2021 in accordance to the increasing trend of recycling waste material.
Your format indication of references in the text is not constant: first you number it (1,2) then you (Author, year)! Why? Please check if you have correct spelling and order of authors names in your references, one of the examples is Ref. 48 – wrong author order.
Response: Thank you. The amendment has been made as shown in the Lines 143, 147, 179, 259, 269, 271, 272, 275, 277, 300, 302, 322, 339, 355, 371, 376, 380, 405, 422, 442 & 451. The author order for Ref. 48 (now Ref. 49) has been updated as shown in the Line 631. - Abstract doesn’t indicate what is novel in your review, why did you wrote it and what you would like to address to a reader.
Response: The abstract has been rewritten by highlighting the novelty of the research and the purpose of this review as shown in the Lines 13 - 28. - Lines 60-61, if you mention several studies, please add also some references, not only one.
Response: Thank you. The reference for that sentence was updated as shown in the Line 78. Furthermore, the overall manuscript was checked to ensure no similar mistake. - Lines 74-87: you have to remove those lines! That is from MDPI template!
Response: Thank you for the kind notification. In the new version of article, we have removed the redundant part as shown from Line 95-104. - The research approach is rather poor described, please check several review papers published lately where they indicated a more detailed info how to performed structured literature search. In your case lines 89-97 are unnecessary, though figure 2 can remain and be updated. It would be more important that you indicate in the beginning of your review in Introduction “WHY did you perform this review and some bullet points indicating it” and based on that structure and novelty to organize your paper.
Response: The research approach has been rewritten as shown in the Lines 106 - 126. The purposes of this review study was shown in the Lines 88 - 94, while the scope of the study was shown in the Lines 123 - 126 (bullet points form). - Conclusions have to be elaborated.
The conclusion has been rewritten with more elaborations on the results as shown in the Lines 467 - 482.
Again, sincerely thanks to the Reviewer for the valuable comments and time. Appreciate it.

Round 2
Reviewer 4 Report
Dear Authors,
Thank you for your elaborated updates. Here will be some comments:
Lines 13-17: Please polish the phrasing for your first sentences in the abstract, make it more readable. It can be written better.
Lines 524-549: Please split your conclusions in several paragraphs or even better shorten and write outcome in a bullet form. Those now are way too long!
Author Response
Dear Honorable Reviewers,
Thanks again for the valuable inputs. The new version of manuscript for second round review is attached below. And here are the highlights for our amendment:
-
Lines 13-17: Please polish the phrasing for your first sentences in the abstract, make it more readable. It can be written better.
Response: Thank you. We have rephrased and shorten it as shown in Lines 13 - 16. -
Lines 524-549: Please split your conclusions in several paragraphs or even better shorten and write outcome in a bullet form. Those now are way too long!
Response: The conclusion is restructured in bullet form as shown in Lines 423 - 448. Perhaps it is more readable.
Once again, appreciate your valuable time in reviewing our manuscript. Thank you.
